# Universal Chromatic Resin-Based Composites: Aging Behavior Quantified by Quasi-Static and Viscoelastic Behavior Analysis

**DOI:** 10.3390/bioengineering9070270

**Published:** 2022-06-22

**Authors:** Nicoleta Ilie

**Affiliations:** Department of Conservative Dentistry and Periodontology, University Hospital, LMU, Goethestr. 70, 80336 Munich, Germany; nilie@dent.med.uni-muenchen.de

**Keywords:** universal chromatic resin-based composite, structural-coloring, viscoelastic, quasi-static, DMA, aging

## Abstract

Universal chromatic dental resin-based composites were recently developed in an attempt to speed up the restoration process with the aim of making it easier for the practitioner to decide on a suitable shade and to avoid time-consuming matching and mixing of materials. The way in which color is created in the analyzed universal chromatic materials is innovative, as it is not only induced by selective light absorption via pigments (Venus Diamond ONE, Venus Pearl ONE), as is usual in regular composites (Charisma Classic, Charisma Topaz, Venus, Venus Diamond), but also by selective light reflection via particularized microstructures (Omnichroma). Material properties were assessed at 24 h post-polymerization and after artificial aging. Flexural strength (n = 20) and modulus were measured in a 3-point-bending test and complemented with fractography and Weibull analysis. Quasi-static (Martens, Vickers, and indentation hardness; elastic and total indentation work; creep, indentation depth) and viscoelastic (storage, loss, and indentation moduli; loss factor) behavior (n = 6) was measured by a depth-sensing indentation test equipped with a DMA module. The nanoscale silica/zirconia polymer core-shell structure in the structural-colored material induces similar or poorer mechanical properties compared with pigment-colored materials, which is related to the higher polymer content. For all materials, aging shows a clear influence on the measured properties, with the degree of degradation depending on the measurement scale.

## 1. Introduction

In most countries, tooth decay poses a major public health challenge and places an enormous burden on the healthcare system [1]. Therefore, modern dental treatment strives to reduce costs and treatment time to make it affordable for everyone, while at the same time being less invasive and aiming to improve quality, durability and aesthetics. In this line, adhesive, tooth-colored restorations used to replace lost tooth tissue and caries defects are the most common interventions [2,3]. Recently, increased efforts to reduce treatment time were noticed. These include applying the restorative material in large increments (bulk-filling) to skip the time-consuming layering process of 2-mm thick increments [4], ultra-fast curing of resin-based composites (RBC) in 3 s coupled with an altering of the polymerization mechanism to a RAFT (reversible-addition-fragmentation chain-transfer) polymerization [5,6], as well as self-adhesive RBCs and derivatives with bioactive and caries-protective abilities [7].

In accordance with the enumerated efforts to speed up the restoration process, universal chromatic RBCs are becoming a valid option entering the portfolio of major dental companies. They have been developed with the aim of making it easier for the practitioner to decide on a suitable shade and to avoid time-consuming matching and mixing of materials with different shades and translucencies [8] in order to achieve the desired aesthetic effect in a clinical situation. More specifically, universal chromatic RBCs were developed to adapt aesthetically to every situation and every patient in order to do justice to the variety of shades of the teeth and restorations [9,10]. An improved blending effect of such RBCs into the environment has indeed been attested [11,12,13], with the caveat, however, that this effect is cavity-size dependent [14,15,16]. The way in which color is created in universal chromatic RBCs is innovative, as it is not only induced by selective light absorption via pigments, as usual in dental materials, but also by selective light reflection via particularized microstructures [17]. The latter is new in dental materials and was inspired by other material science areas [18], while maintaining well known chemical compositions of the organic and inorganic components. Structural-induced coloring is often related to a nanoscale core-shell structure, which consists of a core encapsulated by a shell in which the core has a higher refractive index than the shell. Different types of materials can make up the core and shell domains, such as metals, polymers, and inorganic solids. The structure, size, and composition of these particles can be easily modified to tailor their mechanical, optical, magnetic, thermal, and electro-optical properties [18]. In dental RBCs, the core-shell structure described by the manufacturer as inorganic nanoparticles (silica, zirconia) surrounded by a polymer layer was confirmed by the literature [19], since a multiscale organization of core-shell nanoparticles arranged equidistantly was evidenced, independently whether they are clustered into round, micrometer-sized fillers with different dimensions and distributions, or dispersed in the polymer matrix. In contrast to structurally colored materials, universal chromatic RBCs in which the color has been induced by pigments appear to have a similar microstructure to regular RBCs, evidencing the common variety of fillers, thus encompassing regular dental glass fillers and pre-polymerized fillers in micrometer sizes [19].

In addition, the absence of pigments in structure-colored RBCs showed no influence on material toxicity compared to pigment-colored RBCs in comparable UDMA (urethane dimethacrylate)-based monomer systems for up to three months [19]. As for the mechanical properties described in the literature for universal chromatic RBCs, these were directly related to the content of organic/inorganic fillers. Based on the same line of reasoning, structurally colored RBCs evidenced lower aging resistance, with literature data showing only short aging times of up to 3 months [20].

Reviewing the literature, it must be noted that studies on a longer aging period and a larger set of tested materials that allow comparison are currently lacking, which limits the interpretation of in vitro data in terms of predicting the clinical behavior of the new material developments. Our goal was, therefore, to close this gap and to analyze the long-term aging behavior of structural- and pigment-colored universal chromatic RBCs by exposing them to clinically relevant storage conditions such as artificial saliva and 37 °C, as well as aging. The selection of suitable light-cured control RBCs was based on diversity in terms of microstructure and chemical composition of all components [21]. Therefore, a range of regular RBCs with great variety in filler type, shape, quantity and chemical composition was selected, including compact or pre-polymerized fillers, together with bis-GMA (bisphenol A glycol dimethacrylate) and a bis-GMA-free organic matrix composition.

The following null hypotheses are formulated: the way in which color is induced in RBCs, does not affect their mechanical behavior at the (a) macroscopic, (b) microscopic (quasi-static and viscoelastic behavior) level or (c) after aging.

## 2. Materials and Methods

Three universal chromatic (O, VDO and VPO) and an additional four regular micro-hybrid RBCs, as summarized in Table 1, were analyzed. After 24 h post-polymerization and storage in distilled water at 37 °C, as required by the current ISO standard (ISO 4049:2019 [22]), a 6-month aging period under clinically relevant storage conditions followed.

A blue light emitting diode curing unit (LCU, Bluephase^®^ Style, Schaan, Liechtenstein) was used for polymerization, with an exposure time of 20 s for all materials. The irradiance of the LCU (1405 mW/cm^2^) was measured with a spectrophotometer (MARC, Managing Accurate Resin Curing) system; Bluelight Analytics Inc., Halifax, NS, Canada), while the LCU light guide was applied directly to the device sensor.

### 2.1. Specimen Preparation

The sample preparation was carried out according to the recommendation of ISO 4049:2019 [22] for the 3-point bending test. For this purpose, 280 specimens (n = 40 per RBC) were prepared by filling a white polyoxymethylene mold with an internal dimension of 2 mm × 2 mm × 18 mm with the RBC paste. The material was then pressed between two glass plates separated from the material by polyacetate films. Polymerization occurred by placing the LCU directly on the polyacetate films, following the protocol described in ISO 4049:2009, for 20 s from above and below, overlapping. After demolding, the specimens were immersed in distilled water at 37 °C for 24 h. Half of the samples of each material (n = 20) underwent a 3-point bend test, while the other half underwent additional aging prior to testing. For this purpose, the test specimens were stored for six months in artificial saliva (pH 6.9; 1000 mL: 1.2 g potassium chloride, 0.84 g sodium chloride, 0.26 g dipotassium phosphate, 0.14 g calcium chloride dehydrate) at 37 °C. The immersion medium was changed weekly during the aging period. All broken surfaces were immediately subjected to fractography analysis under a stereomicroscope (Stemi 508, Carl Zeiss AG, Oberkochen, Germany) to determine fracture pattern and fracture origin, and were photographed using a microscope extension camera (Axiocam 305 color, Carl Zeiss AG, Oberkochen, Germany). The origin of fracture was identified as either volume (sub-surface) or surface (edge, corner) defects. Volume (sub-surface) defects could be pores, either inherent in the unpolymerized material or induced during application of the material to the mold to form the beam specimens, or structural inhomogeneity. The mixing of multi-component materials may not be perfectly homogeneous, so that the microstructure contains areas with locally anomalous compositional concentrations, e.g., filler agglomerates or matrix-rich areas, as well as weakly bounded pre-polymer fillers. Surface defects, in contrast, can result from the processing of the RBC sample, since abrasive processing of the sample surface is necessary. In addition, porosities present in the material or formed when the RBC paste is pressed in the mold may also be located on the surface or be connected to a large natural pore or porous area.

### 2.2. Three-Point Bending Test

The flexural strength (FS) and flexural modulus (E) were determined in a 3-point bending test according to NIST No.4877, considering a span of 12 mm [23]. The samples prepared and stored as above were loaded in a universal testing machine until fracture (Z 2.5 Zwick/Roell, Ulm, Germany) at a crosshead speed of 0.5 mm/min. The force during bending was measured as a function of beam deflection, while the slope of the linear part of this curve was used to calculate the flexural modulus.

### 2.3. Instrumented Indentation Test (IIT): Quasi-Static Approach (ISO 14577)

For each RBC and storage conditions, six fragments resulting from the 3-point bend test were selected. The first polymerized side of the specimen was wet-ground with silicon carbide paper (grit size p1200, p2500, and p4000, LECO Corporation, St. Joseph, MI, USA) and polished with a diamond suspension (mean grain size: 1 µm) until the surface was shiny (automatic grinding machine EXAKT 400CS Micro Grinding System EXAKT Technologies Inc., Oklahoma City, OK, USA). A weight of 200 g was used during grinding and polishing. By means of an automated micro-indenter (FISCHERSCOPE^®^ HM2000, Helmut Fischer, Sindelfingen, Germany) equipped with a Vickers diamond tip, one indentation was performed in each specimen. Indentation depth and indentation force were recorded during the whole indentation cycle with the test force increasing from 0.4 mN to 1000 mN, followed by a holding time of 5 s at maximum force and a subsequent reduction in force within 20 s at a constant speed. Change in indentation depth recorded during holding time served as a measure for the material’s creep (Cr). The integral of the indentation force over the depth defines the mechanical work W_total_ (=∫Fdh) which is partially consumed as plastic deformation work W_plast_, while the rest is released as work of the elastic recovery W_elastic_. Indentation creates an impression with the projected indenter contact area (Ac) determined from the force-indentation depth curve, considering the indenter correction based on the Oliver and Pharr model and described in ISO 14577 [24]. The indenter area function was, therefore, calibrated to two different materials with uniform and known material properties (sapphire and quartz glass). Corrections obtained from the tip calibration were then used for further computational data analysis. The indentation hardness (H_IT_ = F_max_/A_c_) is a measure of the resistance to plastic deformation and is convertible to the more familiar Vickers hardness (HV = 0.0945 × H_IT_). The universal hardness (or Martens hardness = F/As(h)) was calculated by dividing the test load by the surface area of the indentation under the applied test load (As), giving a characterization of both plastic and elastic deformation. The indentation modulus (E_IT_) was calculated from the slope of the tangent of indentation depth-curve at maximum force.

### 2.4. Instrumented Indentation Test (IIT); Dynamic Mechanical Analysis (DMA)

The DMA test used a low-magnitude oscillating force (10 different frequencies in the range 0.5–5 Hz) that was superimposed onto a quasi-static force of 1000 mN. The oscillation amplitude was set at five nm, so that the sample deformation stayed within the linear viscoelastic regime. Three randomly chosen indentations were performed per each specimen (n = 6), amounting to 18 individual measurement indentations per RBC brand and aging condition, while ten repeated measurements were performed per each frequency and indentation.

For the used frequency, the force oscillation generates oscillations on the displacement signal with a phase angle δ. The sinusoidal response signal was then separated into a real part and an imaginary part representing the storage (E′) and the loss moduli (E″), respectively. E′ is a measure of the elastic response of a material behavior, whereas E″ characterizes the viscous material behavior. The quotient E″/E′ is defined as the loss factor (tan δ) and is a measure of the material damping behavior.

### 2.5. Statistical Analysis

All variables were normally distributed, allowing a parametric approach to be used. Multifactor analysis of variance was applied to compare the parameters of interest (flexural strength FS, flexural modulus E, Martens, Vickers, and indentation hardness; elastic and total indentation work; creep, indentation depth; storage, loss, and indentation moduli; loss factor). The results were compared using one and multiple-way analysis of variance (ANOVA) and Tukey honestly significant difference (HSD) post hoc test using an alpha risk set at 5%. A multivariate analysis (general linear model) assessed the effect of parameters RBC, aging, and frequency, as well as their interaction terms on the analyzed properties. The partial eta-squared statistic reported the practical significance of each term, based on the ratio of the variation attributed to the effect. Larger values of partial eta-squared (η*P*^2^) indicate a greater amount of variation accounted for by the model (SPSS Inc. Version 27.0, Chicago, IL, USA).

FS data were additionally described by a Weibull analysis. A common empirical expression for the cumulative probability of failure *P* at applied stress σ is the Weibull model [25]:(1)$$P_{f}(\sigma_{c})=1-\exp\left[-{\left(\frac{\sigma_{c}}{\sigma_{0}}\right)}^{m}\right]$$
where $\sigma_{c}$ is the measured strength, *m* the Weibull modulus and $\sigma_{0}$ the characteristic strength, defined as the uniform stress at which the probability of failure is 0.63. The double logarithm of this expression gives: $\ln\ln\frac{1}{1-P}=m\ln\sigma_{c}-m\ln\sigma_{0}$. By plotting ln ln(1/(1 − *P*)) versus ln $\sigma_{c}$, a straight line results, with the upward gradient *m*, whereas the intersection with the x-axis gives the logarithm of the characteristic strength [25].

## 3. Results

### 3.1. Three-Point Bending Test

The parameters measured in the 3-point bending test are illustrated in Figure 1, Figure 2, Figure 3, Figure 4 and Figure 5 and Table 2. A multifactorial analysis shows no influence of aging on flexural modulus (*p* = 0.518) and only a small influence on flexural strength (*p* < 0.001; η*P*^2^ = 0.123). In contrast, the *RBC* exerted a high influence on both flexural strength (η*P*^2^ = 0.764) and modulus (η*P*^2^ = 0.629).

One-way ANOVA clearly distinguishes flexural strength data into three homogeneous subgroups for unaged samples that are related in the following material sequence: (VPO, CT, VD) > (CC, VDO) > (V, O). The flexural modulus allows for a more nuanced distinction between materials, classified into four homogeneous subgroups in descending order: (VPO, VD) > (VPO, CC) > (CC, CT, VDO) > (V, O). After aging, the analyzed materials were distributed into five homogeneous subgroups (VDO, VD) > (VDO, CT) > (CT, VPO) > (VPO, CC) > (V, O) with regard to flexural strength and in four homogeneous subgroups with regard to flexural modulus: VDO > (VD, CT, VPO) > (CC, V) > (O).

For unaged specimens (Figure 3a), the highest percentage of failures came from sub-surface defects (61.4%), followed by edges (21.4%) and corners (17.1%). After 6 months of storage in artificial saliva (Figure 3b), fractures caused by edge defects predominated (66.4%), followed by corners (23.6%), while only 10% of fractures were caused by volume defects (sub-surface).

Specific modes of failure impart characteristic features on the fracture surface, as exemplified in Figure 4 on some representative images evidencing a fracture originated from (a) the corner, (b) sub-surface and (c) an edge defect.

The reliability of the flexural strength for analyzed materials and aging conditions was evaluated by a Weibull statistic (Figure 5, Table 2). Within the 95% confidence interval, the limits of which were calculated by subtracting and adding 1.96 times the standard error of the mean (Table 2), the material reliability expressed by the Weibull parameter m was highest at VPO both 24 h post-polymerization and after aging. During aging, material reliability is slightly degraded (O, CC, VPO), while the ranking of materials within an aging condition changes only slightly (Table 2). High R^2^ values are observed for all groups, indicating a good fit of the Weibull model to the measured data.

### 3.2. Instrumented Indentation Test (IIT): Quasi-Static Approach

A multifactorial analysis indicates a significant (*p* < 0.001) and very strong effect of the RBC on all measured quasi-static parameters in descending order of influence: W_t_ (η*P*^2^ = 0.987), W_e_ (η*P*^2^ = 0.976), E_IT_ (η*P*^2^ = 0.961), HM (η*P*^2^ = 0.923), h_max_ (η*P*^2^ = 0.923), HV (η*P*^2^ = 0.881) and Cr (η*P*^2^ = 0.881). The effect of aging is comparably strong for W_e_ (η*P*^2^ = 0.976) and W_t_ (η*P*^2^ = 0.942), while it consistently decreases for the other parameters related to the influence of RBC: h_max_ (η*P*^2^ = 0.891), E_IT_ (η*P*^2^ = 0.540), HM (η*P*^2^ = 0.310), HV (η*P*^2^ = 0.218), Cr (η*P*^2^ = 0.10) (Figure 6, Figure 7 and Figure 8).

### 3.3. Instrumented Indentation Test (IIT); Dynamic Mechanical Analysis (DMA)

A multifactorial analysis shows that all three analyzed factors—RBC, aging and frequency—exerted a significant (*p* < 0.01) influence on the measured parameters. The stronger effect from all three factors was observed for the RBCs, while the strongest influence of the RBC was observed on indentation and storage modulus (η*P*^2^ = 0.944), followed closely by H_IT_ (η*P*^2^ = 0.937), then the loss factor (η*P*^2^ = 0.715) and, finally, the loss modulus (η*P*^2^ = 0.656). As the second parameter in terms of strength of influence on the measured properties, frequency had the strongest influence on the viscoelastic parameter loss modulus (η*P*^2^ = 0.693) and loss factor (η*P*^2^ = 0.902), followed by indentation (η*P*^2^ = 0.795) and storage modulus (η*P*^2^ = 0.791), while the effect on the indentation hardness was smaller (η*P*^2^ = 0.262). Finally, aging exerted a moderate influence on storage modulus (η*P*^2^ = 0.454), indentation modulus (η*P*^2^ = 0.453) and indentation hardness (η*P*^2^ = 0.4369), while the effect on the viscoelastic parameters was extremely small, at the limit of the significance level: loss factor (η*P*^2^ = 0.089), and the loss modulus (η*P*^2^ = 0.003).

The variation with frequency for H_IT_ was small (Figure 9), with a similar pattern of variation in all materials and a slight increase up to 1.1 Hz. VD and VDO showed statistically similar and the highest H_IT_ values in both unaged and aged samples, while the values decreased significantly during aging in all materials. The group of VPO and CT behaved similarly for unaged samples and followed the group with the highest hardness values for unaged samples. They then differentiate after aging, with CT showing slightly but significantly higher values. The lowest H_IT_ values were recorded for O in unaged samples. The differences to the next higher H_IT_ values (V) are reduced after aging.

The pattern of variation in indentation modulus with frequency (Figure 10) is similar for all materials and aging conditions, with the highest values being shown at the lowest frequencies, followed by a steady decrease up to 1.4 Hz. The material ranking is comparable to the hardness and all values decrease with aging. The VDO and VD and CT and VPO groups behave similarly in both the unaged and aged states, while the moduli measured in V and O were the lowest.

The loss factor (Figure 11) also shows a comparable pattern for all materials in its dependence on frequency, with an inverse ordering of the materials compared with the indentation modulus and an increase for all materials after aging.

## 4. Discussion

The continuous development of dental materials, and especially their advertising as new material categories, requires a thorough characterization of properties and behavior and, in particular, a direct comparison with materials that have already been clinically proven. Elaborate comparisons of material properties measured in vitro with their in vivo performance when placed in a dental cavity are quite sobering, since a large number of parameters are required to predict clinical behavior, which cannot be reduced to a few key laboratory tests [26]. With this mentioned limitation, however, it must be emphasized that, albeit weakly to moderately, some parameters could be filtered out as they clearly indicate a correlation with clinical behavior. This includes parameters selected for this study, such as the flexural strength and modulus, as well as the hardness [26]. In contrast, the measurement of the viscoelastic behavior of dental materials is a relatively new field that does not yet allow a correlation with clinical behavior. Parameters, such as the loss factor, which characterize the damping behavior of a material, can be linked, albeit only intuitively at present, to a failure mechanism of brittle materials that is important in a clinical situation, namely chipping.

The present study chose for comparison for the new, clinically not yet proven universal chromatic RBCs, their regular shaded version, when available. The materials VDO and VD, as well as VPO and CT, are obviously very similar, if not identical, with regard to chemical composition and filler content (Table 1). With a few exceptions at the limit of statistical significance, these materials behave similarly in terms of the measured properties and their aging behavior. The highest inorganic filler amount observed in VD and VDO relates to the highest mechanical properties measured at both micro and macroscopic scales. VD and VDO differ significantly from the other materials in the DMA measured IIT parameters, while in the macroscopically measured strength the group competes with CT and VPO. This different behavior when evaluating properties on different scales is related to the microstructure, since it has been shown that the filler size and the extent of failure caused by surface defects at three-point bending correlate. Because fillers can be pulled out during grinding, while creating surface depressions large enough to initiate fracture, the larger filler sizes observed in VD (20 µm) compared with CT (5 µm) [27] may account for larger surface defects able to initiate failure and, thus, reduce flexural strength. This argument is strengthened by the observation that in the nanostructured material O, failure was predominantly triggered by intrinsic sub-surface defects rather than surface defects. The particular microstructure of this material, which exhibits nanoscale core-shell structures connected by the polymer shell, allows for a much smoother surface after grinding without pulling out particles that can cause large defects. However, the mechanical properties in O were the lowest among the materials studied, which must be related to the lower proportion of inorganic fillers, since the polymer shell that is part of the filler system is also organic in nature.

Since fracture is considered to be the primary cause of failure of RBC restorations clinically [28], macroscopic strength measurements and an assessment of the cause of failure are essential. In this context, determining the reliability of a material plays a very important role, since the strength measured in the laboratory can be transferred to the much smaller dimensions of a filling using the size–strength relationship defined by Weibull when the effective volume or surface is known [29,30]. A two-parameter Weibull model is frequently used to characterize brittle dental materials [30] and describes the reliability of a material depending on the scatter of its defects. The minimum number of samples of 20 for an unbiased result of the Weibull analysis was met in the present study [31], while the Weibull model was well described by the measured data, which is evidenced by the high R^2^ values summarized in Table 2. In contrast to the micro-scaled measured IIT-parameters, the aging effect in the macroscopic test was smaller, or even not significant, as underlined by the statistical multifactorial analysis of variance and the lower partial eta-squared values. However, the three-point bending measurements cannot be considered less important in predicting the aging effect, as this was clearly reflected in a reduced Weibull modulus in all materials tested (Table 2). This observation reinforces the need to carry out a Weibull analysis for a deeper evaluation of strength data.

As can be seen from Figure 4, the fracture in the analyzed materials originated either from surface or volume (sub-surface) defects in the tensile zone of the beam subjected to three-point bending. While volume defects are most often associated with a large natural pore or porous area, which may either already be present in the material or may have been introduced when the RBC paste was condensed into the mold to form the samples, defects intrinsic to the microstructure may also be seen as a potential fracture origin. Particularly for materials that have pre-polymer filler content such as CT and VPO, the weaker bonding of the pre-polymer filler to the matrix can be considered a cause of failure or stress riser [27]. The same applies to agglomerates or compositional inhomogeneity. Interestingly, there is a clear shift from sub-surface (volume) to predominantly surface-initiated fracture during aging, indicating surface weakening, which is also reflected in significantly lower micromechanical properties but not necessarily lower macro mechanical properties. Figure 4 also shows how a crack propagates from the fracture origin when subjected to a specific stress in the tensile region where the local stress concentration is greatest. The crack propagates radially, although not necessarily uniformly radially, resulting in some asymmetry in the microscale appearance of the fracture surface near the origin.

Since hydrolytic degradation is one of the main disadvantages of methacrylate-based RBC restorations [32] and has been observed in various matrix formulations [33], in vitro aging under clinically simulated conditions is essential to characterize a novel material. As described in Table 1, the organic matrix of the analyzed RBCs is based on dimethacrylates, including monomers such as bis-GMA, UDMA, and TEGDMA. These monomers contain a variety of chemical bonds such as esters, ethers, urethanes and amides that can be cleaved by hydrolysis [34]. Mechanical properties such as strength and durability were shown to decrease due to degradation [34], which is also confirmed by the cured study, with the observation that degradation was most evident in microscale measured properties, particularly in DMA analysis. The latter showed a reduction in hardness and indentation modulus, which was corroborated with an increase in loss factor and loss modulus, as a result of plasticization and degradation in all materials analyzed, regardless of matrix chemistry. In addition to the hydrolytic degradation of functional groups in the monomers, the connection between the filler and the matrix via amphiphilic silanes with ester groups in the methacrylate part and siloxane groups in the alkoxy part, which tends to degrade, also offers another possibility for degradation [35].

The DMA analysis encompasses frequencies relevant to human chewing activity, which was estimated between 0.94 Hz and 2.17 Hz [36]. Interestingly, while the indentation hardness increases slightly in this frequency segment, the indentation modulus decreases for all materials examined, which indicates a less good adaptation to the oral environment. The behavior can be explained by the fact that the time that the polymer matrix has to adapt to the load decreases with increasing frequency [37]. As is known for polymer-based composites, viscoelastic material behavior occurs by stress dissipation due to friction at the interphase boundary [38]. While microstructural peculiarities play an important role in this aspect, the complex filler–matrix interface in such highly filled materials does not allow a clear statement due to different filler sizes and distribution. Rather, properties, such as damping behavior, are related to the organic matrix content. The structure colored material shows the highest damping capacity, followed by V. The high damping capacity observed in V, in spite of the low polymer content, must be related to the polymer matrix as it is the only material analyzed that does not contain UDMA or UDMA derivatives in its composition. The latter leads to a polymer matrix with a higher modulus of elasticity compared with the bis-GMA systems used in V. In addition, an increased occurrence of voids in the fractured surfaces of V was observed, which can also lead to increased stress relaxation. Furthermore, lower cross-linking can also lead to increased damping as it is less restrictive for the thermal movements of molecular chains [37]. On a general note, a clear inverse relationship was observed between damping on the one hand and strength, modulus of elasticity and hardness on the other.

Therefore, all null hypotheses can be rejected. The properties measured in universal chromatic RBCs are directly related to the amount of filler and organic matrix, rather than the way color was induced, and are within the range of analyzed clinically successful materials.

## 5. Conclusions

Aging shows a clear influence on the measured properties in the analyzed RBCs, whereby the strength of the degradation depends on the observation scale. While quasi-static and DMA IIT-parameters are more sensitive in detecting changes in material properties, a three-point bend test may also be considered a sensitive predictor, provided it is complemented by a Weibull analysis.

Universal chromatic RBCs, either structurally or pigment colored, behave in the range of clinically successful RBCs with regard to the measured properties and aging behavior, with measured properties being related to the inorganic filler content. A comparable clinical behavior is to be expected.

## Figures and Tables

**Figure 1 bioengineering-09-00270-f001:**
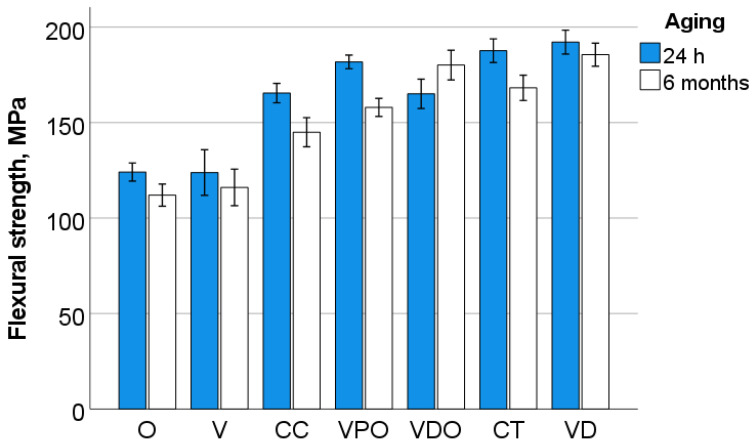
Flexural strength as a function of RBC and aging (mean values with 95% confidence interval).

**Figure 2 bioengineering-09-00270-f002:**
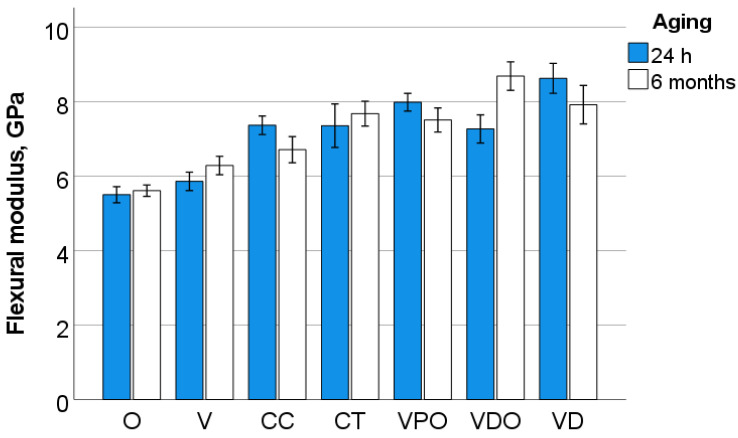
Flexural modulus as a function of RBC and aging (mean values with 95% confidence interval).

**Figure 3 bioengineering-09-00270-f003:**
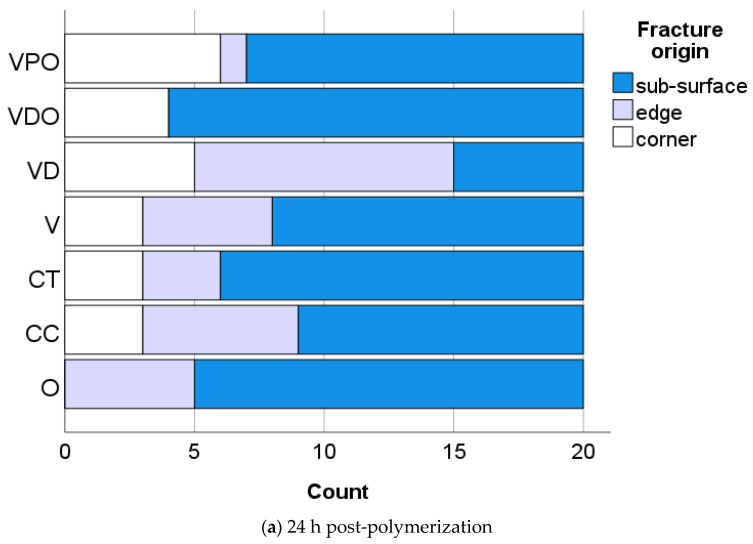
Fracture origin as a function of RBC and aging.

**Figure 4 bioengineering-09-00270-f004:**
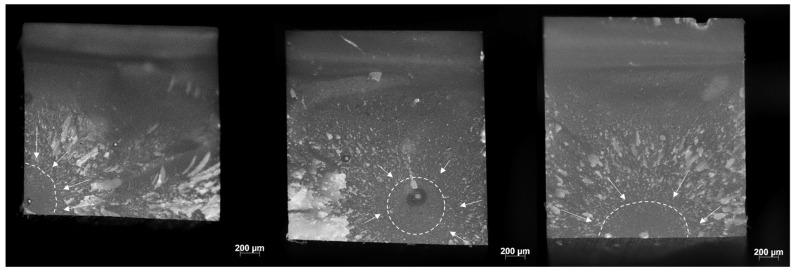
Representative images of the three identified fracture origins in the analyzed materials, from left to right: corner (surface), sub-surface (volume), and edge defect (surface). The fracture mirror (smooth surface in the initial part of the fracture created when the crack is accelerated) is marked by a dashed line, with arrows pointing to the origin of the fracture. The rougher surface adjacent to the mirror (mist) is followed by crack propagation in different directions, resulting in radial striations (hackle lines).

**Figure 5 bioengineering-09-00270-f005:**
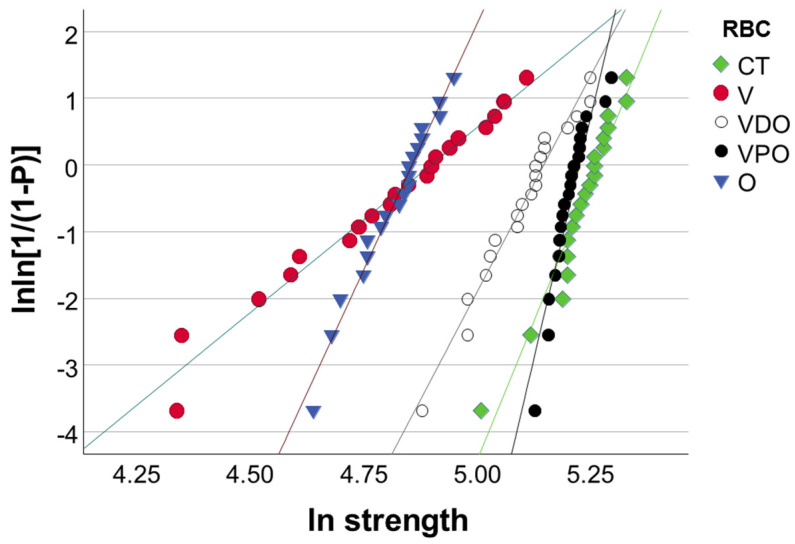
Example of a Weibull plot representing the empirical cumulative distribution function of strength data at 24 h post-curing. Linear regression was used to numerically assess the goodness-of-fit and estimate the parameters of the Weibull distribution, as described in Table 2.

**Figure 6 bioengineering-09-00270-f006:**
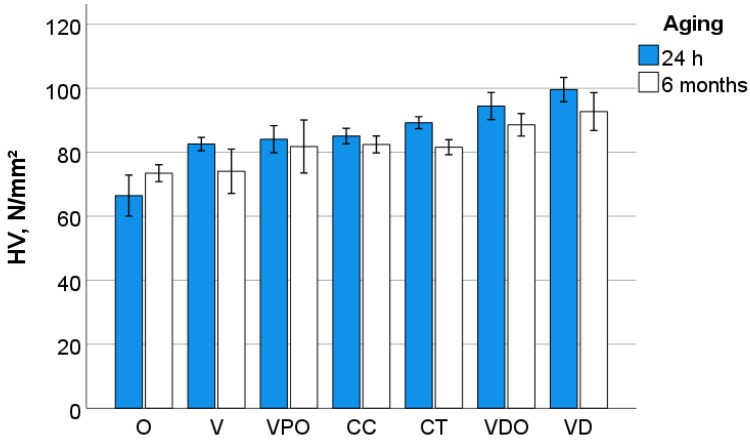
Variation of the Vickers hardness (HV) as a function of RBC and aging conditions (mean values with 95% confidence interval).

**Figure 7 bioengineering-09-00270-f007:**
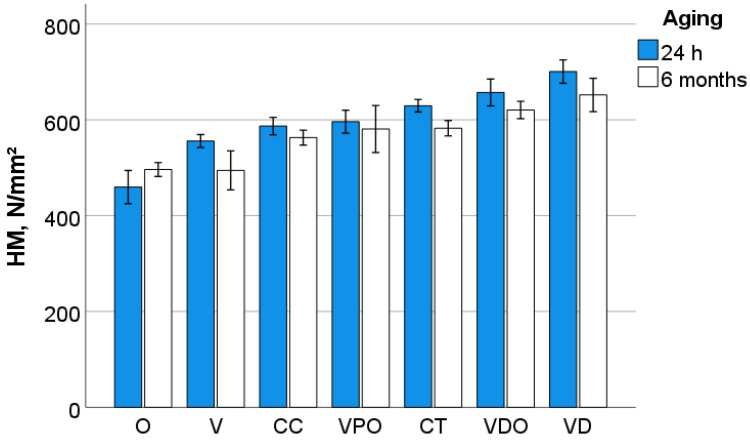
Variation of the Martens hardness (HM) as a function of RBC and aging conditions (mean values with 95% confidence interval).

**Figure 8 bioengineering-09-00270-f008:**
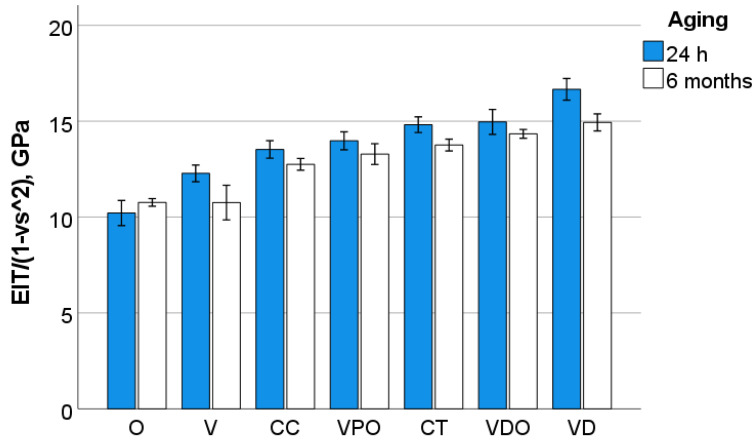
Variation of the indentation modulus as a function of RBC and aging conditions (mean values with 95% confidence interval).

**Figure 9 bioengineering-09-00270-f009:**
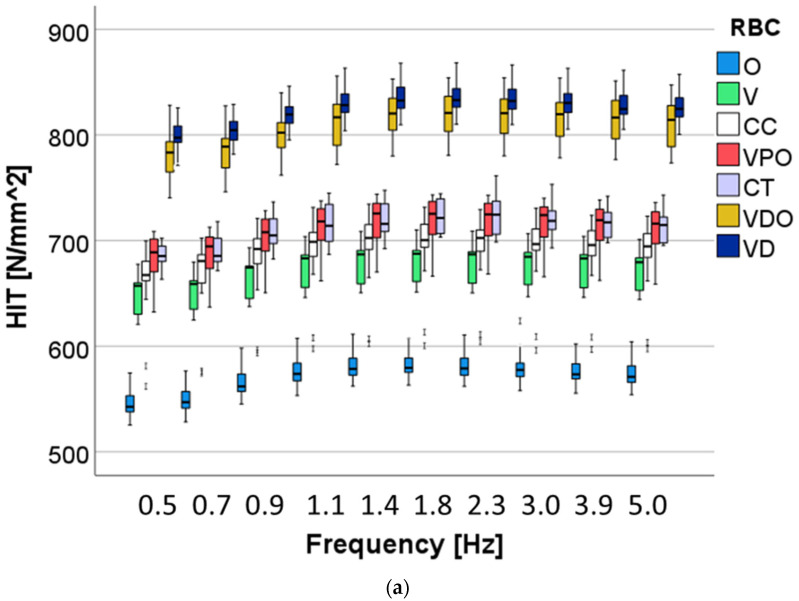
Variation of the indentation hardness as a function of material and frequency in (**a**) unaged and (**b**) after 6 months’ immersion in artificial saliva.

**Figure 10 bioengineering-09-00270-f010:**
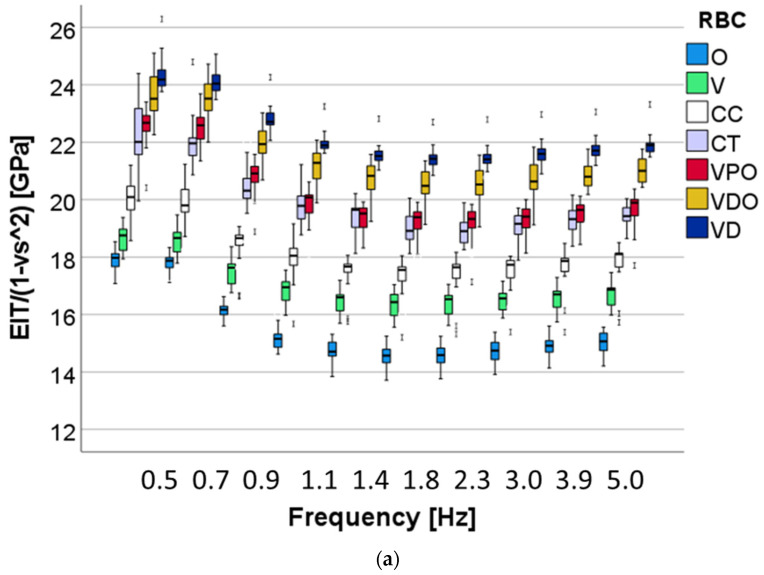
Variation of the indentation modulus as a function of material and frequency in (**a**) unaged and (**b**) after 6 months’ immersion in artificial saliva.

**Figure 11 bioengineering-09-00270-f011:**
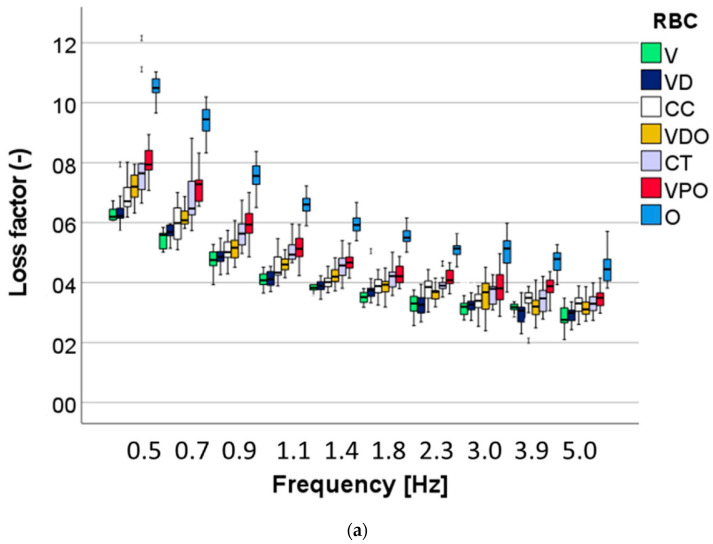
Viscoelastic material behavior: Variation of the loss factor (tan delta) as a function of material and frequency in (**a**) unaged and (**b**) after 6 months’ immersion in artificial saliva.

**Table 1 bioengineering-09-00270-t001:** Analyzed RBCs and chemical composition as indicated by the manufacturer.

Code	RBC/Manufacturer	Shade	LOT	Monomer	Filler	
					Composition	wt/vol%
**CC**	Charisma Classic Kulzer	A4	K010710	bis-GMA, UDMA, TEGDMA	Ba−Al−B−F−Si glass, SiO_2,_ Tectosilicate, PPF	77/61_t_(60)_i_
**CT**	Charisma Topaz Kulzer	CL	K010028	UDMA, TCD-DI-HEA, TEGDMA	Ba−Al−B−F−Si glass, PPF, SiO_2_	75/59_t_(58)_i_
**O**	Omnichroma Tokuyama	one	0551	UDMA, TEGDMA	SiO_2_, ZrO_2_ and CF	79/68
**V**	Venus Kulzer	A1	K010518	bis-GMA, TEGDMA	Ba−Al−B−F−Si glass, SiO_2_	78/59
**VD**	Venus Diamond Kulzer	YO	K010035	UDMA, TCD-DI-HEA, TEGDMA	Ba−Al−B−F−Si glass, SiO_2_	81/64
**VDO**	Venus Diamond ONE Kulzer	one	VP2210	UDMA, TCD-DI-HEA, TEGDMA	Ba−Al−B−F−Si glass, SiO_2_	81/64
**VPO**	Venus Pearl ONE Kulzer	one	VP211019	UDMA, TCD-DI-HEA, TEGDMA	Ba−Al−B−F−Si glass, PPF, SiO_2_	75/59_t_(58)_i_

Abbreviations: bis-GMA = bisphenol A glycol dimethacrylate; TEGDMA = triethylene glycol dimethacrylate; UDMA = urethane dimethacrylate; TCD-DI-HEA = 2-propenoic acid; (octahydro-4,7-methano-1*H*-indene-5-diyl) bis(methyleneiminocarbonyloxy-2,1-ethanediyl) ester; PPF = pre-polymerized filler; SiO_2_ = silicon oxide (silica); ZrO_2_ = zirconium oxide; BaO−Al_2_O_3_−SiO_2_ = barium aluminosilicate glass; B_2_O_3_−F−Al_2_O_3_−SiO_2_ = boroaluminosilicate.

**Table 2 bioengineering-09-00270-t002:** Weibull parameters arranged in ascending order of Weibull parameter m (with standard error SE) within each aging category. R squared (R^2^) values are also given.

Aging	RBC	R^2^	m	SE
24 h	V	0.98	5.55	0.21
	VDO	0.97	12.93	0.52
	O	0.98	14.81	0.53
	CT	0.95	16.64	0.93
	VD	0.97	17.45	0.67
	CC	0.96	18.26	0.86
	VPO	0.89	28.99	2.40
6 months	V	0.92	6.39	0.44
	O	0.87	9.42	0.85
	CC	0.96	10.66	0.52
	VDO	0.96	13.04	0.66
	CT	0.96	14.32	0.71
	VD	0.97	17.07	0.66
	VPO	0.97	19.24	0.78

## Data Availability

Data are available on requests.

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
