# Peer review of "Universal Chromatic Resin-Based Composites: Aging Behavior Quantified by Quasi-Static and Viscoelastic Behavior Analysis"

_bioengineering, 2022, doi:10.3390/bioengineering9070270_

Round 1

Reviewer 1 Report

Dear authors,

the article covers a very interesting topic and I support its publication.

There is kind of a tendency to self-citation (9 articles !). By the way I suggest some minor adjustements to improve the article for the readers.

In the introduction the authors wrote:

“They have been developed with the aim of making it easier for the practitioner to decide on a suitable shade and to avoid time-consuming matching and mixing of materials with different shades and translucencies in order to achieve the desired aesthetic effect in a clinical situation.“

This sentence could be supported by a reference about the role of a translucency in the selection of a resin-based composite.

Among the possible papers I suggest: PMID: 23390620 

“however with the amendment that this effect is cavity size dependent [12-14] “

I suggest to add the reference  PMID: 26835524.  This could help the readers to understand alternative ways to deal with big and high c-factor cavities with single-shade universal composites.

Author Response

All comments to the corresponding author have been addressed independently below. The authors’ rebuttal is always in BLUE and where changes have been added to the revised manuscript in light of the reviewer comments these are presented in RED.

The author would firstly like to thank the reviewers for taking the time to read and critically appraise the manuscript and secondly to thank the reviewers’ for their positive constructive comments in improving the work.

Comments and Suggestions for Authors

Reviewer comment: Dear authors, the article covers a very interesting topic and I support its publication.

Author’s response:  we are grateful for this appreciation.

There is kind of a tendency to self-citation (9 articles !). By the way I suggest some minor adjustements to improve the article for the readers.

Author’s response:  Thank you for this relevant note.  Unfortunately, I did not noticed that a working group was cited so often. This fact arose because the topic of universal chromatic materials is new and the mentioned group has been intensively and significantly involved in it in terms of material characterization.  The above citations have been shortened accordingly.

Reviewer comment: In the introduction the authors wrote: “They have been developed with the aim of making it easier for the practitioner to decide on a suitable shade and to avoid time-consuming matching and mixing of materials with different shades and translucencies in order to achieve the desired aesthetic effect in a clinical situation.“ This sentence could be supported by a reference about the role of a translucency in the selection of a resin-based composite.

Among the possible papers I suggest: PMID: 23390620 

Author’s response:  thanks for the comment; I accept your suggestion and have cited the paper mentioned above.

Reviewer comment: “however with the amendment that this effect is cavity size dependent [12-14] “ I suggest to add the reference  PMID: 26835524.  This could help the readers to understand alternative ways to deal with big and high c-factor cavities with single-shade universal composites.

Author’s response:  thank you for this observation, however, the three cited publications deal precisely with the topic of universal chromatic resin composites and the matching of their color to the environment, depending on the cavity size, compared to regular materials, and under clinical conditions.

Reviewer 2 Report

Dear Author,

I have been invited to review your work entitled “Universal chromatic resin-based composites: aging behavior quantified by quasi-static and viscoelastic behavior analysis”. I congratulate with you for this work of concern. Here are few notes to improve the manuscript: 

Abstract

The results should be clearer and a short conclusion should be written also in the abstract.

Introduction

  • In the first lines, a reference should be added.

Materials and methods/Results

  • This is a complex article. I suggest simplifying the procedures and data presentation. Moreover, statistically significant differences should be presented clearer.

Author Response

All comments to the corresponding author have been addressed independently below. The authors’ rebuttal is always in BLUE and where changes have been added to the revised manuscript in light of the reviewer comments these are presented in RED.

The author would firstly like to thank the reviewers’ for taking the time to read and critically appraise the manuscript and secondly to thank the reviewers’ for their positive constructive comments in improving the work.

Comments and Suggestions for Authors

Reviewer comment: I have been invited to review your work entitled “Universal chromatic resin-based composites: aging behavior quantified by quasi-static and viscoelastic behavior analysis”. I congratulate with you for this work of concern.

Author’s response:  I am grateful for this appreciation.

Here are few notes to improve the manuscript: 

Abstract

The results should be clearer and a short conclusion should be written also in the abstract.

Author’s response:  the abstract has been edited – please take into account changes in the revision.  Please also note the style of the abstract, which differs from regular dental journals and does not contain values ​​of measured properties, but rather important trends in the results and brief descriptions.

Introduction

  • In the first lines, a reference should be added.

Author’s response:  thank you for this very important observation. I added in the revision the required reference: “Hugo, F.N.; Kassebaum, N.J.; Marcenes, W.; Bernabé, E. Role of Dentistry in Global Health: Challenges and Research Priorities. J Dent Res 2021, 100, 681-685, doi:10.1177/0022034521992011.

Materials and methods/Results

  • This is a complex article. I suggest simplifying the procedures and data presentation. Moreover, statistically significant differences should be presented clearer.

Author’s response:  I am grateful for this comment. There is indeed a lot of data, which is necessary for the complex characterization at different measurement scales. Also, please keep in mind that the readers of the journal are not necessarily dentists and require a complex description of the methods used, the mathematics behind them, and well-defined statistics, including Weibull and multi-factorial analysis. The methods are presented individually and the statistics correspond to today's procedures and requirements.  I have tried to present everything as clearly and concisely as possible to the best of my knowledge and belief.

Reviewer 3 Report

Dear author,

thank you very much for the opportunity to review this in general nicely written manuscript dealing with an interesting new material class. From my point of view, there are some points that need to be adapted as listed below: 

Abstract: 

Please be more specific in the Abstract regarding the M&M section. You should name the used materials, methods, specimen numbers and point out some important results using numerical data. For that, you can shorten the rationale a bit. 

Introduction:

The introduction is well written and organized. However, the term core-shell structure should be defined early in the introduction, as it appears several times. At the beginning of the introduction, some more references would be nice, as the global burden study (Kassebaum et al) to point out the demand for dental restorative materials. Furthermore, the introduction could be shortened a bit, as there are some paragraphs that belond rather to the discussion part ("the hypothesis that the absence...)

As you refer to a null-hypothesis at the end of the discussion, please rephrase the hypotheses as null-hypotheses to point out the aims of this work. 

M&M:

Please provide the working distance for polymerization. 

Table 1: To reviewer it is unclear, why Charisma Classic is written in bold letters. 

Please provide a clear defintion for volume and surface defects.

2.3: Which aspect of the fragments was polished? Which pressure was applied during polishing?

Results:

Figures: The materials used should appear in the same sequence as in the M&M section. Furthermore, I would suggest to indicate in the graphs, which materials are conventional and which are universal chromatic. This facilitates reading.

Discussion: 

I would suggest to structure the whole manuscript divided in conventional materials on one side and new universal chromatic materials on the other side. The discussion needs to focus more on the meaning of corner, edge or volume defects. Please focus more on the reasons, why you would expect different mechanical properties and different aging behaviour caused by the different structure of the new material class. 

"Since fracture is considered to be the primary cause of failure of RBC restorationsclinically" I would be more specific and name the, from my point of view more common, other failurereasons as secondary caries or adhesion loss. 

Please use the terms "in vitro" or "in vivo" whith uniform formatting and italic, whenever they appear. 

"a less good adaptation to the oral environmental." Should say "oral environment" instead. 

"The latter lead..." should say "The latter leads". Please double-check grammar and english language. 

Null-Hypotheses: Contradictory to the author, I believe that none of the hypotheses could be rejected, as the materials performed very similar. As mentioned beofre, I would suggest to rephrase the hypotheses. 

Author Response

All comments to the corresponding author have been addressed independently below. The authors’ rebuttal is always in BLUE and where changes have been added to the revised manuscript in light of the reviewer comments these are presented in RED.

The author would firstly like to thank the reviewers for taking the time to read and critically appraise the manuscript and secondly to thank the reviewers for their positive constructive comments in improving the work.

Comments and Suggestions for Authors

Reviewer comment: Dear author, thank you very much for the opportunity to review this in general nicely written manuscript dealing with an interesting new material class.

Author’s response:  I am grateful for this appreciation.

Reviewer comment: From my point of view, there are some points that need to be adapted as listed below: Abstract: Please be more specific in the Abstract regarding the M&M section. You should name the used materials, methods, specimen numbers and point out some important results using numerical data. For that, you can shorten the rationale a bit. 

Author’s response:  thanks for the pertinent comment – your suggestions have been implemented within the limits of the space and journal regulations. The used materials, measured parameters, and specimen numbers have been added. Numerical data however is barely not possible in such short abstracts, considering the high number of measured parameters, the number of materials analyzed, and two different storage conditions. Therefore, general information was used to indicate the trends and the main results.

Reviewer comment: Introduction: The introduction is well written and organized. However, the term core-shell structure should be defined early in the introduction, as it appears several times. At the beginning of the introduction, some more references would be nice, as the global burden study (Kassebaum et al) to point out the demand for dental restorative materials. Furthermore, the introduction could be shortened a bit, as there are some paragraphs that belond rather to the discussion part ("the hypothesis that the absence...)

Author’s response:  thank you for the comment – indeed the global burden study was mentioned here – thanks for pointing this out. It is now added in the revision. In addition, I reduced the introduction by reducing the mentioned paragraph. It is not part of the discussion in my view as what has been published on the materials so far is summarized as a preamble to what is analyzed in the present work.  However, I agree that the way I wrote this paragraph, e.g.  "The hypothesis that... was not confirmed" was a bit muddled and actually sounds like a discussion, so I've shortened and rephrased it as recommended. There is also a cited reference for the core-shell structure and an expansion for clarity, please consider marked modification during the manuscript.

Reviewer comment: As you refer to a null-hypothesis at the end of the discussion, please rephrase the hypotheses as null-hypotheses to point out the aims of this work. 

Author’s response:  In fact that A and B perform similarly, as in the original text, is a null hypothesis. However, I have rephrased them in a simpler way: “The following null hypotheses are formulated the way in which color is induced in RBCs, does not affect their mechanical behavior at a) macroscopic, b) microscopic (quasi-static and viscoelastic behavior) level or c) after aging

M&M: Please provide the working distance for polymerization. 

Author’s response:  thank you for this observation, the missing info was added in “specimen preparation”.

Table 1: To reviewer it is unclear, why Charisma Classic is written in bold letters. 

Author’s response:  My apologies for this error - it resulted while translating the original manuscript into the journal format.

Reviewer comment: Please provide a clear defintion for volume and surface defects.

Author’s response:  I added explanations to what surface and volume defects are – please consider the addition in the chapter “Specimen preparation”.

Reviewer comment: 2.3: Which aspect of the fragments was polished? Which pressure was applied during polishing?

Author’s response:  Specimens have been light-cured from top and bottom, as stipulated in ISO 4049  and indicated in the manuscript, therefore the aspect of the fragments is irrelevant. Nevertheless, I worked systematically, and have always used the first light-cured side. This information was now provided in the manuscript “For each RBCs and storage condition, six fragments resulting from the three-point bend test were selected for each material and storage condition. The first polymerized side of the specimen was…”

As for the pressure – the specimens were wet-ground and then polished in an automatic grinding machine. This way, additional pressure can be exerted by using weights.  In this case, it was 200 g.  The information has now been added to the manuscript text.

Reviewer comment: Results: Figures: The materials used should appear in the same sequence as in the M&M section. Furthermore, I would suggest to indicate in the graphs, which materials are conventional and which are universal chromatic. This facilitates reading.

Author’s response:  in MM, the materials are presented in alphabetical order. In the results, the materials are deliberately presented in ascending order of measured properties. This is the recommended way of presenting data in science to more logically track trends and impacts.

Reviewer comment: Discussion: I would suggest to structure the whole manuscript divided in conventional materials on one side and new universal chromatic materials on the other side. The discussion needs to focus more on the meaning of corner, edge or volume defects. Please focus more on the reasons, why you would expect different mechanical properties and different aging behaviour caused by the different structure of the new material class. 

Author’s response:  Well, universal chromatic materials are not a new material class, as was intended shown in the present paper. They are newly introduced materials that we don't know that much about at the moment and are struggling with a lot of publicity. Universal chromatic materials are in fact, based on their microstructure, chemistry, and main compounds, dental RBCs. Only Omnichroma has a new micro-structure – the core-shell structure (new for dental material not for material science).  Therefore, a comparison in material categories makes no sense. This applies to all material class comparisons.  Instead, restorative materials must be characterized and analyzed individually and their properties are not related to the material category they are advertised to belong to, as clearly stated in the literature. While I am a strong proponent of fractography, which is also apparent in the way the research in this paper was conducted and the detailed explanations presented in the discussion, it was not the intention of the paper to identify the origin of fracture precisely. Light microscopy allows to determine the location of the defects, however, whether it is an agglomerate, a debonding, a pore, a void, a pre-polymer filler, etc, can be determined, and not always, by examining each of the 280 specimens by SEM at high magnification. That was beyond the scope of the present investigation.

Reviewer comment: "Since fracture is considered to be the primary cause of failure of RBC restorationsclinically" I would be more specific and name the, from my point of view more common, other failurereasons as secondary caries or adhesion loss. 

Author’s response:  I do agree that there is also another reason for the failure of RBCs restorations, but the literature is very clearly showing that in RBCs the primary cause of failure is fracture.

Reviewer comment: Please use the terms "in vitro" or "in vivo" whith uniform formatting and italic, whenever they appear. 

Author’s response:  thanks for this observation – I carefully checked the terms and made the corrections.

Reviewer comment: "a less good adaptation to the oral environmental." Should say "oral environment" instead. 

Author’s response:  OK

Reviewer comment: "The latter lead..." should say "The latter leads". Please double-check grammar and english language. 

 Author’s response:  OK, thanks. Grammar and English were checked.

Reviewer comment: Null-Hypotheses: Contradictory to the author, I believe that none of the hypotheses could be rejected, as the materials performed very similar. As mentioned beofre, I would suggest to rephrase the hypotheses. 

Author’s response:  I strongly disagree with the reviewer's comments that the materials performed very similarly. The statistical analysis is clearly showing that RBC is the main factor of influence. Their mechanical properties varied greatly and were clearly related to the inorganic filler content. The same applies to their behavior after aging. Please take a  look at the results again.

I have been invited to review your work entitled “Universal chromatic resin-based composites: aging behavior quantified by quasi-static and viscoelastic behavior analysis”. I congratulate with you for this work of concern.

Author’s response:  I am grateful for this appreciation and would like to thank you for the invested time in reviewing this paper.

Round 2

Reviewer 2 Report

Dear Author, 

Thank you for providing the revised version of your work. The manuscript was improved and I suggest publishing it on Bioengineering.

Thank you for your hard work. 

Author Response

Thank you for the time invested and the pertinent comments.

Reviewer 3 Report

Dear author, 

thank you of the adaptation of the manuscript! There are only two minor points that need to be adapted:

Introduction:

Structural-induced coloring is often related to a nanoscale core-shell structure which consists of a core encapsulated by a shellin which the core has a higher refractive index than the shell.

Reviewer Comment: Can you give an exemplary information about the content of the core and the shell, respectively? 

Results:

Regarding Author’s response:  I strongly disagree with the reviewer's comments that the materials performed very similarly. The statistical analysis is clearly showing that RBC is the main factor of influence. Their mechanical properties varied greatly and were clearly related to the inorganic filler content. The same applies to their behavior after aging. Please take a  look at the results again.

Reviewer Comment: To point out the differences between the materials more clearly, you should add some symbols indicating significant differences to all figures.

Author Response

All comments to the corresponding author regarding the second revision have been addressed independently below.

Comments and Suggestions for Authors

Reviewer comment:

Introduction:

Structural-induced coloring is often related to a nanoscale core-shell structure which consists of a core encapsulated by a shell in which the core has a higher refractive index than the shell.

Reviewer Comment: Can you give an exemplary information about the content of the core and the shell, respectively? 

Author’s response:  I have expanded the core-shell composition information, as suggested.  Please note the change in the manuscript.  As this is a technically very complex and large area, a few examples will not suffice.  Please also note that the composition of the core-shell structure used in the dental material has been explained in detail (silica and zirconia coated with polymer).

Reviewer comment:

Results:

Regarding Author’s response:  I strongly disagree with the reviewer's comments that the materials performed very similarly. The statistical analysis is clearly showing that RBC is the main factor of influence. Their mechanical properties varied greatly and were clearly related to the inorganic filler content. The same applies to their behavior after aging. Please take a  look at the results again.

Reviewer Comment: To point out the differences between the materials more clearly, you should add some symbols indicating significant differences to all figures.

Author’s response:  Please note that the statistical analysis is multifactorial. Therefore, this complex statistic cannot be indicated by symbols in a graphical representation. The results chapter summarizes the eta-squared values for each factor of influence to show the effect size of each parameter. The effect size of "material" was the strongest.